# Correlation of Expression Changes between Genes Controlling 5-HT Synthesis and Genes *Crh* and *Trh* in the Midbrain Raphe Nuclei of Chronically Aggressive and Defeated Male Mice

**DOI:** 10.3390/genes12111811

**Published:** 2021-11-18

**Authors:** Olga E. Redina, Vladimir N. Babenko, Dmitry A. Smagin, Irina L. Kovalenko, Anna G. Galyamina, Natalia N. Kudryavtseva

**Affiliations:** 1FRC Institute of Cytology and Genetics, Siberian Branch of Russian Academy of Sciences, 630090 Novosibirsk, Russia; bob@bionet.nsc.ru (V.N.B.); smagin@bionet.nsc.ru (D.A.S.); koir@bionet.nsc.ru (I.L.K.); galyamina@bionet.nsc.ru (A.G.G.); n.n.kudryavtseva@gmail.com (N.N.K.); 2Pavlov Institute of Physiology, Russian Academy of Sciences, 199034 Saint Petersburg, Russia

**Keywords:** midbrain raphe nuclei, transcriptome, serotonin, corticotropin-releasing hormone, thyrotropin-releasing hormone, behavior, chronic social conflict

## Abstract

Midbrain raphe nuclei (MRNs) contain a large number of serotonergic neurons associated with the regulation of numerous types of psychoemotional states and physiological processes. The aim of this work was to study alterations of the MRN transcriptome in mice with prolonged positive or negative fighting experience and to identify key gene networks associated with the regulation of serotonergic system functioning. Numerous genes underwent alterations of transcription in the MRNs of male mice that either manifested aggression or experienced social defeat in daily agonistic interactions. The expression of the *Tph2* gene encoding the rate-limiting enzyme of the serotonin synthesis pathway correlated with the expression of many genes, 31 of which were common between aggressive and defeated mice and were downregulated in the MRNs of mice of both experimental groups. Among these common differentially expressed genes (DEGs), there were genes associated with behavior, learning, memory, and synaptic signaling. These results suggested that, in the MRNs of the mice, the transcriptome changes associated with serotonergic regulation of various processes are similar between the two groups (aggressive and defeated). In the MRNs, more DEGs correlating with *Tph2* expression were found in defeated mice than in the winners, which is probably a consequence of deeper *Tph2* downregulation in the losers. It was shown for the first time that, in both groups of experimental mice, the changes in the transcription of genes controlling the synthesis and transport of serotonin directly correlate with the expression of genes *Crh* and *Trh*, which control the synthesis of corticotrophin- and thyrotropin-releasing hormones. Our findings indicate that CRH and TRH locally produced in MRNs are related to serotonergic regulation of brain processes during a chronic social conflict.

## 1. Introduction

The serotonergic system is an important neurotransmitter system that is involved in the brain-driven regulation of various behaviors and physiological, metabolic, hormonal, and psychoemotional processes [1,2,3,4] as well as in the development of affective and neurological disorders (reviewed in refs. [5,6,7,8,9,10]). Nonetheless, due to the multifunctionality of serotonin, the question of the specificity of its role has always been raised, because alterations in the activity of the serotonergic system may be either a consequence of serotonin-driven regulation of neurobiological processes or a symptom of a developing disease.

The effect of serotonin neurotransmission on social cognition and emotion management can be modulated by both genotype [11,12] and other factors, such as gender [13], age [14], as well as environmental factors [15,16], which makes it difficult to study the basic molecular genetic mechanisms that determine the synthesis of serotonin and its role in the formation of the phenotype on the human population. Despite efforts to study genetic control of serotonergic regulation of various processes in the brain, this field remains largely unexplored.

Midbrain raphe nuclei (MRNs) are some of the most relevant brain structures for research on the mechanisms underlying serotonergic regulation of behavior because this brain area contains a large number of serotonergic neurons [1,17] and has interconnections with multiple brain regions. Thus, this brain area has been chosen to investigate the neurogenomic pathways of regulation of serotonergic activity in MRNs of mice in the sensory contact model described in [18,19]. The essence of this model is that, under the conditions of a chronic social conflict, aggressive mice with prolonged positive fighting experience (winners) and defeated mice with negative fighting experience (losers) develop various pathological forms of behavior. Male mice exhibiting repeated aggression in daily agonistic interactions enter a pathological psychotic-like state characterized, e.g., by symptoms of increased aggressiveness, hyperactivity, stereotypical behavior, and disturbances in social recognition (reviewed in [20,21]). Mice with chronic social defeat experience develop a mixed anxiety/depression-like state with symptoms of hypoactivity, behavioral deficits, anhedonia, and other problems [22,23,24,25,26]. We have previously demonstrated that, in the MRNs of both groups of animals with the opposite social experiences, there is underexpression of several key genes responsible for the synthesis, transport, and binding of serotonin [27,28,29].

Recently, next-generation sequencing showed high efficiency in the elucidation (in cells and tissues) of the molecular genetic mechanisms underlying the development of various pathologies. Earlier, using high-throughput RNA sequencing (RNA-Seq), we have found that the induction of aggressive or depressive behavior in mice in the model of sensory contact is associated with changes in the expression of many genes in the ventral tegmental area (VTA) [30]. In that study, a comprehensive analysis of these shifts in transcriptomes made it possible to identify the genes most likely associated with the specific features of VTA activity during the development of opposite types of behavior in experimental animals [30].

The aim of the present investigation was to study in detail the features of transcriptome changes in the MRNs of mice that had repeated experiences of either victories or defeats (winners and losers) in daily agonistic interactions. As mentioned above, this area of the brain is suitable for research on expression alterations of genes participating in the regulation of the serotonergic system and other systems functionally related to it under the influence of various external and/or endogenous factors that affect behavior. Therefore, most attention in this article is given to genes associated with the functioning of the serotonin system in the MRNs of mice under the conditions of chronic social conflict.

## 2. Materials and Methods

### 2.1. Animals

The experiments were carried out using 10–12 week-old C57BL/6J male mice. The mice were kept in the Conventional Vivarium (federal research center Institute of Cytology and Genetics, SB RAS, Novosibirsk, Russia) under standard conditions at 22 ± 2 °C on a 12/12 h light–dark cycle (lights on at 8:00 AM) with dry laboratory feed and water available ad libitum. All procedures were conducted in compliance with European Communities Council Directive 210/63/EU of 22 September 2010. The study protocol was approved by Scientific Council No. 9 of the Institute of Cytology and Genetics SB RAS of 24 March 2010, No. 613 (Novosibirsk, Russia). 

### 2.2. Induction of Opposite Types of Social Behavior in Mice by Agonistic Interactions

The opposite types of social behavior were induced by means of daily agonistic interactions (intermale confrontations) of mice as described previously [19,22]. Pairs of weight-matched mice were each placed in a cage (28 × 14 × 10 cm) bisected by a perforated transparent partition allowing the animals to see, hear, and smell each other but preventing physical contact. The animals were left undisturbed for 2 or 3 days to adapt to the unfamiliar housing conditions before they were exposed to encounters. Every day at 14:00–17:00 PM (Russian local time), the cage lid was replaced by a transparent one, and after 5 min (the period necessary to activate agonistic interactions), the partition was removed for 10 min for the intermale confrontation. The mouse that attacked, bit, and chased the opponent was considered the winner. The superiority of the winner was established according to the outcome of two or three encounters with the same opponent. A mouse that showed only defensive behavior (sideways postures, upright postures, withdrawal, lying on the back, or freezing) was defined as the loser. If the aggressive attacks were very active and long, then the interactions between the mice were stopped after 3 min (or even earlier) by re-insertion of the partition to prevent injuries in the defeated mouse (only males were studied). This means that the painful effects of agonistic interactions are absent in defeated mice in this model.

Each defeated mouse (loser) was exposed to the same winner for 3 days, and then, to continue the agonistic interactions, the defeated mouse was placed in an unfamiliar cage with an unfamiliar winner behind the partition. Each winner remained in its original cage. The intermale confrontation procedure was performed once a day for 21 days and yielded equal numbers of winners and losers.

Three groups of animals were set up: (1) controls, i.e., mice without a daily experience of agonistic interactions; (2) winners, i.e., a group of aggressive mice chronically winning during 21 days in the daily agonistic interactions (intermale confrontations); and (3) losers, i.e., mice with chronic experience of defeats during 21 days in the daily agonistic interactions. These animals with the opposite types of social experience developed various pathological behaviors. Mice with a prolonged experience of aggression and victories (winners) are known to exhibit increased aggressiveness, hyperactivity, stereotypical behaviors, anxiety, impaired social recognition, irritability, autistic spectrum symptoms, a condition similar to drug addiction, and other problems (reviewed in [20]). The losers manifest mixed anxiety/depression-like behaviors accompanied by full immobility, avoidance of any social interactions, helplessness, indifference, and other aberrations [25,31].

Twenty-one pairs of C57BL/6J mice were employed to induce the opposite types of social behavior. Winners and losers with the most pronounced behavioral phenotypes were chosen for transcriptome analysis. The following criteria were used. To be designated as losers, during the activation period (5 min before a fight), the mice had to demonstrate all symptoms of depressive behavior: to not approach the partition, to sit in the cage corner opposite to the partition or with the nose into a corner or litter; the mice had to exhibit immobility, freezing during a winner’s attack, or indifference in all experimental situations (without behavioral reactions); no inversions of behavior to the opposite one after a change of aggressors; and avoidance and passive defense when attacked by the aggressor. To be designated as winners, during the activation period, the mice had to demonstrate strong aggressive motivation, and every day, they had to attack the opponent immediately after partition removal, stopping only for rest and to display manic motivation to bite the opponent in spite of full submission.

The control animals and all experimental mice were decapitated simultaneously. Experimental mice were decapitated 24 h after the last agonistic interaction. The brain regions were dissected by the same experimenter according to a relevant map in the Allen Mouse Brain Atlas [32]. All biological samples were placed in the RNAlater solution (Thermo Fisher Scientific, Waltham, MA, USA) and stored at −70 °C until sequencing.

### 2.3. RNA-Seq

The frozen MRN samples from the mice (winners (*n* = 3), losers (*n* = 3), and controls (*n* = 3)) were sent to JSC Genoanalytica (Moscow, Russia), which specializes in RNA-Seq. For transcriptome profiling, mRNA was extracted by means of the Dynabeads mRNA Purification Kit (Ambion, Thermo Fisher Scientific, Waltham, MA, USA). cDNA libraries were constructed using the NEBNext mRNA Library Prep Reagent Set for Illumina (NEB, Ipswich, MA, USA). Single-end sequencing of cDNA libraries was performed on the Illumina HiSeq 1500 platform (Illumina Sequencing, San Diego, CA, USA) with a read length of 50 bases. All samples were analyzed as three biological replicates. Quality metrics of the mapped data (Appendix A) were determined in the Spliced Transcripts Alignment to a Reference (STAR) software [33]. The sequencing data were mapped to the mouse reference genome (GRCm38.p3) available in GenBank using the TopHat2 aligner (Center for Bioinformatics and Computational Biology, University of Maryland, College Park, MD, USA) [34]. The RNA-Seq datasets are available in the European Nucleotide Archive (Accession No. PRJEB47635).

Cufflinks/Cuffdiff programs were utilized to estimate gene expression levels in fragments per kilobase of transcript per million mapped reads (FPKM) and to identify differentially expressed genes (DEGs) in the experimental groups relative to the control. Genes were considered differentially expressed at a false discovery rate (q value) <5% [35]. Only annotated gene sequences were included in the analysis.

### 2.4. Validation of RNA-Seq Data

We have previously conducted studies on gene expression in males in similar experiments by the real-time PCR method with a larger number of samples in each experimental group, i.e., winners and losers (>10 animals). In comparison with the control, the direction and extent of changes in the expression of *Tph2*, *Slc6a4*, and several other genes in the MRNs of males in these experimental groups are generally consistent between the two utilized methods: real-time PCR [27,28] and RNA-Seq [29]. To cross-validate our results, we also employed a unique resource from Stanford University, USA [36], and noted significant concordance with our RNA-Seq data pool [37]. These findings suggested that the transcriptome analyses of the data provided by JSC Genoanalytica were valid, and that the method reflects the actual processes that occur in the brain under our experimental paradigm.

### 2.5. Functional Annotation of DEGs

This procedure was performed using the DAVID (database for annotation, visualization, and integrated discovery) gene annotation tool [38]. The *Mus musculus* genome served as the background list for over-representation analysis. The gene ontology (GO) option in DAVID as well as Kyoto Encyclopedia of Genes and Genomes (KEGG) pathway database [39] were used to identify significantly (*p* < 0.05) enriched biological processes and metabolic pathways. To determine the association of DEGs with a behavior/neurological phenotype, the Neurological Disease Portal (Phenotypes, Mouse) in the rat genome database [40] was employed. An atlas of combinatorial transcriptional regulation in mice and humans [41] was used to identify DEGs encoding transcription factor genes.

### 2.6. Statistical Methods

The acquired RNA-Seq data (in FPKM values) were log-transformed, centered, and normalized. The normalized RNA-Seq data were subjected to correlation analysis. A 99% and 99.9% (df = 4; *p* < 0.01 (0.917); *p* < 0.001 (0.974)) two-tailed confidence interval was considered significant. Software packages Statistica 6.0 (StatSoft, Tulsa, OK, USA) and JACOBI4 [42] were used for data analysis and presentation.

## 3. Results

### 3.1. DEGs in the MRNs of Winners vs. Control Mice

A total of 14474 expressed genes were detected. Among them, 348 genes were found to be differentially expressed in the MRNs of aggressive mice (winners) compared to controls (Appendix A). Of these, 276 genes (79.3%) had a lower transcription level in the winners than in controls. Functional annotation of the DEGs revealed 94 genes associated with a behavior/neurological phenotype, among which there were genes associated with abnormal emotion/affect behavior, abnormal aggression-related behavior, increased aggression towards mice, abnormal depression-related behavior, and abnormal fear/anxiety-related behavior (Appendix A).

Among the DEGs, there were 34 genes encoding transcription factors, 11 of which are associated with a behavior/neurological phenotype (Appendix A). Functional annotation in the KEGG database allowed us to identify the metabolic pathways most significantly enriched in the DEG set (Table 1). A detailed description of the DEGs related to these pathways is given in Appendix A. Most of the DEGs assigned to the listed metabolic pathways had a lower transcription level in the winners than in the control animals.

### 3.2. DEGs in the MRNs of Losers vs. Controls

A total of 14445 expressed genes were detected, among which 214 genes were found to be differentially expressed in the MRNs of the defeated mice (losers) compared to the controls (Appendix A). Most of these DEGs, namely 185 genes (86.4%), had a lower level of transcription in the MRNs of the defeated mice as compared to the controls. Altogether, 76 DEGs proved to be associated with a behavior/neurological phenotype, among which there were genes related to abnormal emotion/affect behavior, abnormal aggression-related behavior, increased aggression towards mice, abnormal depression-related behavior, and abnormal fear/anxiety-related behavior (Appendix A). Twenty-two DEGs encode transcription factors, and among them, 10 genes are associated with a behavior/neurological phenotype. Table 2 presents the metabolic pathways most significantly enriched in this DEG set. A detailed description of the DEGs related to these pathways is given in Appendix A. Most of the DEGs assigned to the listed metabolic pathways have a lower transcription level in the defeated mice than in the control animals.

Most of the metabolic pathways thus identified in the MRNs of the defeated mice were similar to those in the aggressive animals. The difference is that, in the MRNs of the defeated mice, DEGs dealing with the cholinergic synapse were found. On the other hand, the terms specific to MRNs of the aggressive mice were ribosome, cardiac muscle contraction, oxytocin signaling pathway, GABAergic synapse, fatty acid biosynthesis, amphetamine addiction, tyrosine metabolism, ECM-receptor interaction, butanoate metabolism, morphine addiction, dopaminergic synapse, and long-term depression.

### 3.3. DEGs Shared by the Winners and Losers (Common DEGs)

A comparison of the DEG lists revealed 158 common genes that manifested significantly altered expression in the MRNs of both the winners and losers as compared to the controls (Appendix A). For all the common DEGs in the MRNs, the transcription level changed unidirectionally between the winners and losers with respect to the controls. One hundred forty-two (89.9%) out of 158 common DEGs were downregulated by the social confrontation. Fifty-five common DEGs are associated with a behavior/neurological phenotype (Appendix A). KEGG analysis indicated that the terms most significantly enriched in the set of common DEGs are MAPK signaling pathway, serotonergic synapse, focal adhesion, calcium signaling pathway, and type II diabetes mellitus (Appendix A). All genes assigned to these metabolic pathways turned out to be significantly downregulated in the MRNs of mice of both experimental groups.

### 3.4. Expression of DEGs Encoding Proteins Responsible for the Synthesis and Transport of Serotonin and Hormones

In MRNs, many neurons are serotonergic. Our data indicated that expression of genes encoding key proteins that control the synthesis (TPH2 and DDC) and transport (reuptake) of serotonin (SLC6A4) was low in the MRNs of mice of both experimental groups (Table 3).

It is known that, when the body is exposed to endogenous or exogenous factors, functional changes in tissues and organs are primarily determined by neurotransmitter systems and through the action of biologically active substances (hormones). According to our data, in the MRNs of both winners and defeated mice that experienced chronic social confrontation for 21 days, there was a highly significant decrease in the expression of genes *Crh* and *Trh* encoding corticotropin-releasing hormone (CRH) and thyrotropin-releasing hormone (TRH) (Table 3).

### 3.5. Expression of Neurotransmitter and Hormone Receptors

Our findings revealed shifts in the expression of several genes encoding neurotransmitter and hormone receptors (Table 4). Readers can see that, under the conditions of social confrontation, the effects of various neurotransmitter and hormonal signals can modulate the functioning of MRNs in mice. The expression of six of the genes presented in Table 4 (*Chrm1*, *Gabra4*, *Hcrtr1*, *Htr5b*, *Irs4*, and *Nr2f2*) significantly changed in the MRNs of both winners and defeated animals under the conditions of chronic social confrontations. The most significant alterations in transcription levels were documented for *Htr5b*, which encodes 5-hydroxytryptamine (serotonin) receptor 5B.

### 3.6. Correlation between mRNA Levels of Genes Encoding Proteins Responsible for the Synthesis and Transport of Serotonin and the Expression of Genes Coding for Hormones or Neurotransmitter or Hormone Receptors

In both experimental groups of mice, changes in the transcription level of genes controlling the synthesis (*Tph2* and *Ddc*) and transport (*Slc6a4*) of serotonin directly and most significantly correlated with the expression of genes encoding serotonin (*Htr5b*) and insulin (*Irs4*) receptors as well as with transcription levels of genes *Crh* and *Trh* encoding corticotropin-releasing hormone and thyrotropin-releasing hormone, respectively (Table 5). There was an inverse correlation with the expression of the *Nr2f2* gene. Expression of DEGs encoding cholinergic receptors and hypocretin (orexin) receptor 1 did not correlate with mRNA levels of genes encoding proteins that control serotonin synthesis and transport. Regarding GABA receptors, in both experimental groups, the *Ddc* transcription level positively correlated with *Gabra4* expression (Table 5).

### 3.7. A Comparison of Molecular Mechanisms Correlating with mRNA Expression of Genes Encoding Proteins That Govern Serotonin Synthesis in the MRNs of Mice with the Opposite Social Experiences in Daily Confrontations

To identify the main features associated with decreased expression of genes involved in serotonin synthesis in the MRNs of aggressive and defeated mice, we analyzed correlations between DEG expression and the expression of *Tph2*, which encodes the rate-limiting enzyme of serotonin biosynthesis.

The results of the correlation analysis showed that the expression of 33 DEGs found in the MRNs of winning mice correlated with *Tph2* expression (Appendix A), whereas in the MRNs of defeated mice, the expression of 98 DEGs correlated with *Tph2* expression (Appendix A). It is important to note that 31 genes are common between these lists. Accordingly, we can say that decreased expression of genes involved in the synthesis and transport of serotonin in the MRNs of mice is a nonspecific mechanism behind the functional response of this brain region to social confrontation. Furthermore, the observed more significant *Tph2* underexpression in the MRNs of the defeated mice probably involves a larger number of genes related to the formation of phenotypic features of the serotonergic-system activity in the MRNs of losers. Biological processes associated with behavior, learning, memory, and synaptic signaling were identified in the analysis of DEGs correlating with *Tph2* expression in the MRNs of both experimental groups of mice. In contrast to the winners, in the MRNs of the losers, changes in processes associated with neurogenesis were present too (Figure 1 and Figure 2).

In the set of the 33 DEGs whose expression correlated with *Tph2* transcription in the MRNs of the winners, the largest number of correlations (33) was found for three genes, *Gch1*, *Slc6a4*, and *Trh*; their expression correlated with all DEGs in this set of genes (Appendix A). In the set of 98 DEGs whose expression correlated with *Tph2* transcription in the MRNs of losers, the largest number of correlations was found for *Tac2*, *A2m*, and *Crh*; their expression levels correlated with almost all DEGs (97, 96, and 96, respectively) in this gene set (Appendix A).

## 4. Discussion

We analyzed the impact of chronic social confrontations on levels of gene transcription in the MRNs of male C57BL/6J mice. It is reported here that, in the MRNs of experimental mice, significant changes in the expression of a large number of genes occur both in winners and in defeated males. During the manifestation of the opposite types of social behavior, transcription levels of some genes are expected to change in opposite directions, as shown earlier in the analysis of the VTA transcriptome of the same groups of animals [30]; however, in the MRNs of winners and defeated mice, no genes were found that changed the level of transcription in a direction opposite to that in the opponent group. Furthermore, in the MRNs, numerous genes (158 DEGs) underwent transcription alterations that were unidirectional between the winners and defeated animals. The MAPK signaling pathway is linked with the regulation of a wide range of cellular processes, including proliferation, differentiation, apoptosis, and stress responses in both normal and pathological conditions [43]; this signaling cascade was identified here as a common and most severely altered metabolic pathway (highly enriched in DEG sets). This finding suggests that mice in both experimental groups were stressed.

MRNs are known as a brain region characterized by high density of serotonergic neurons, the functioning of which is associated with the regulation of behavior and psychoemotional states [44]. According to our functional annotation in KEGG, there was significant downregulation of several genes related to the serotonergic synapse metabolic pathway in the MRNs of mice of both experimental groups; these can be considered key genes in this work because their products control the serotonin synthesis (*Tph2* and *Ddc*), transport (*Slc6a4*), and binding (*Htr5b*). In earlier reports from our group, the decrease in mRNA levels of genes encoding the proteins that manage the synthesis and transport of serotonin in the MRNs of mice that had opposite social experiences has already been discussed in detail, and the reduced transcription has been confirmed by real-time PCR [27,28]. The objectives of the present project were to identify additional most significant molecular determinants of serotonergic regulation of aggressive and submissive behaviors. We assumed that the genes whose expression would significantly correlate with the mRNA levels of genes encoding the proteins controlling the synthesis and transport of serotonin would be highly likely associated with serotonergic signaling.

Several serotonin receptors are expressed in MRNs; however, here, significant changes in the transcription level were documented only for two of them: *Htr3a* and *Htr5b*. The expression of *Htr3a* in the MRNs was statistically significantly lower only in defeated mice, and diminished *Htr5b* transcription in the MRNs was registered in both experimental groups of mice.

Recently, it was shown that *Htr3a* (encoding 5-hydroxytryptamine (serotonin) receptor 3A) is implicated in the regulation of goal-oriented behavior via the serotonergic projection from the median raphe nucleus to ventral hippocampus [45]. By contrast, in our study, *Htr3a* expression did not correlate with the expression of genes encoding the proteins responsible for serotonin synthesis.

*Htr5b* codes for 5-hydroxytryptamine (serotonin) receptor 5B, which is expressed in serotonergic neurons both in the dorsal raphe nucleus (DRN) and median raphe nucleus [46], where it functions as an autoreceptor [47], i.e., is localized on the presynaptic membrane of the neuron and binds a specific ligand released by that same neuron, thereby implementing a feedback mechanism for monitoring the neurotransmitter synthesis and/or release.

*Htr5b* expression highly positively correlated here with mRNA levels of genes encoding the proteins that manage serotonin synthesis (*Tph2* and *Ddc*) and transport (*Slc6a4*). Additionally, it was shown for the first time that the expression of *Htr5b*, *Tph2*, *Ddc*, and *Slc6a4* highly positively correlates with the expression of genes *Crh* and *Trh*. According to results of single-cell RNA-Seq analysis of serotonergic neurons in the murine DRN and median raphe nucleus, the genes encoding both neuropeptides (CRH and TRH) are expressed predominantly in neurons of the DRN [46].

The contribution of CRH concentration changes in raphe nuclei to the regulation of behavior and psychoemotional states is poorly understood at present. Some experimental evidence that alterations in CRH levels in the DRN may mediate stress-related and emotional/affective phenomena has been obtained and reviewed [48]. There are two known genes (*Crhr1* and *Crhr2*) coding for CRH receptors, and the interaction of CRH with either receptor can trigger sensitization of DRN neurons thereby leading to a subsequent greater release of serotonin in response to CRH input; this phenomenon may, at least in part, be responsible for the behavioral aberrations associated with depression or anxiety [48]. Microinjection of CRH into the DRN has revealed that the interaction of CRH with these receptors can modulate behavioral consequences of uncontrolled stress in a dose-dependent manner. It has been suggested that low doses of injected CRH preferentially bind to CRHR1 and inhibit DRN 5-HT activity, whereas higher doses of CRH are expected to bind to both receptor subtypes and to no longer inhibit DRN serotonin activity. CRHR2 is thought to mediate excitation of DRN 5-HT neurons [49]. Even though CRH is presumably expressed predominantly in DRN cells [46], a number of studies also point to an important role of the median raphe nucleus in CRH-related signaling, which contributes to behavior regulation. It has been reported that CRH injection into the median raphe nucleus significantly elevates memory-dependent fear expression in rats [50]. The results of another experiment showed that the increased medial-prefrontal-cortex serotonin release caused by the infusion of CRH into the DRN can be abrogated by inactivation of the median raphe nucleus [51]. It was concluded in that research article that neurons of both the DRN and median raphe nucleus are involved in complex CRH-driven modulation of the serotonergic activity in the medial prefrontal cortex [51].

In our study, *Crhr1* expression did not change under the influence of social confrontations, whereas *Crhr2* expression was not detectable in the MRNs of mice of both experimental groups. We did not find a significant correlation between expression levels of *Crh* and *Crhr1*; however, we did detect a significant positive correspondence of expression levels between *Crh* and *Crhbp* (encoding CRH-binding protein (Table 6), which regulates CRH bioavailability) [52,53]. These findings suggest that, in our experiment, CRH signaling and its possible effect on behavior are primarily associated not with receptors CRHR1 and CRHR2 but rather with the level of *Crhbp* expression. In support of this hypothesis, it is possible to cite a number of studies indicating an important function of CRHBP in the regulation of behavior. Genotypic analysis suggests that a single-nucleotide substitution in the *CRHBP* gene may be related to a decrease in aggressive behavior [54]. In another study, genotype analysis revealed a positive correlation between the *CRHBP* gene and a suicidal tendency [55]. Our results support a recently advanced hypothesis that not only the CRH synthesized in the hypothalamus but also the CRH locally produced in CRH-containing raphe nuclei in the brainstem is important for stress adaptation [56].

It has been repeatedly demonstrated that thyroid hormones can serve as an effective adjunctive treatment of affective disorders [57]. A review article on a modulatory impact of exogenous thyroid hormones on the serotonin system of the brain in affective disorders indicates that raphe nuclei participate in these processes [58]. As for TRH, which is endogenously synthesized in MRNs [46], its role in the regulation of serotonin synthesis and behavior has hardly been studied.

Table 6 shows that, in our work, *Trh* expression did not correlate with the expression of its receptors (*Trhr* and *Trhr2*) and thyrotropin-releasing-hormone-degrading enzyme (*Trhde*) in the MRNs. Nevertheless, *Trh* mRNA expression corresponded to mRNA levels of genes encoding the proteins that control the synthesis and transport of serotonin as well as to the expression of the *Irs4* gene encoding insulin receptor substrate 4, which is known as an adapter molecule involved in the signal transduction of both insulin [59] and leptin [60]. Mutations in *IRS4* are associated with central hypothyroidism [61], and the TSH response to the TRH test is blunted in male *IRS4* mutation carriers having central hypothyroidism [62]. This evidence suggests that IRS4 is required for proper TRH signaling.

The expression of the genes discussed above positively correlated here with the expression of genes encoding proteins that control the synthesis and transport of serotonin and negatively correlated with *Nr2f2* expression in the MRNs of mice of both experimental groups. Nuclear receptor subfamily 2 group F member 2 gene (*Nr2f2*, also known as *COUP-TFII*) codes for a member of the steroid/thyroid hormone superfamily of nuclear receptors. NR2F2 is a ligand-activated transcription factor exerting complex pleiotropic effects on glucose homeostasis, insulin sensitivity, and lipid metabolism in various tissues [63,64]. It is reported that insulin represses *NR2F2* gene expression in pancreatic β-cells [65], and that the regulation of *NR2F2* promoter activity is cell type specific [66,67]. Nevertheless, on the basis of our data, we can assume that insulin can play an important part in the modulation of biological processes in the MRNs of mice of both experimental groups, and that its influence on the level of *Nr2f2* transcription can be regulated through IRS4. Moreover, it is known that *Nr2f2* belongs to the group of genes whose transcriptional activation is associated with all-*trans*-retinoic acid [66,68]. In early studies, it was reported that COUP-TF dimers bind to different GGTCA response elements, allowing COUP-TF to repress hormonal induction of vitamin D3, thyroid hormone, and retinoic acid receptors [69]. Therefore, even though specific features of the regulation of NR2F2 activity in raphe nuclei have not yet been studied, our data are in good agreement with what was described above about this regulation in other tissues.

Our results indicate mRNA underexpression of TPH2 (the rate-limiting enzyme of brain serotonin synthesis) and of DDC and SLC6A4 (proteins controlling serotonin synthesis and transport) in the MRNs of both winners and defeated mice. Additionally, it turned out that the expression of a large number of DEGs in the MRNs in both experimental groups correlates with *Tph2* expression. Moreover, 31 genes were common between these lists and the changes in their transcription were unidirectional between the lists. Accordingly, our data imply that a decrease in the synthesis and transport of serotonin in murine MRNs is a nonspecific mechanism behind the cellular response to chronic social confrontation.

## 5. Conclusions

Our results mean that chronic social confrontations significantly affect transcription levels of numerous genes in the MRNs of both aggressive and defeated mice. mRNA underexpression of key genes encoding the proteins that control the synthesis of serotonin (TPH2 and DDC) and transport (reuptake) of serotonin (SLC6A4) is demonstrated in the MRNs of mice of both experimental groups. Data analysis suggests that the reduction in serotonin synthesis in the MRNs of the winners and losers as a consequence of chronic social confrontation is nonspecific, because *Tph2* expression correlated with the expression of many common genes whose MRN transcription levels shifted in the same direction between the experimental groups. Among these DEGs, genes associated with behavior, learning, memory, and synaptic signaling were identified. A larger number of DEGs whose expression correlated with the *Tph2* expression level was found in the MRNs of the defeated mice, which can probably be explained by a more substantial decrease in *Tph2* transcription in the MRNs of the losers.

Here, it was shown for the first time that, in both groups of experimental mice, alterations in expression of genes encoding the proteins that govern the synthesis and transport of serotonin directly correlated with expression of genes *Htr5b*, *Crh*, *Trh*, and *Irs4*. An inverse correlation was observed with the expression of *Nr2f2*. We can hypothesize that, along with genes encoding the proteins that manage the synthesis and transport of serotonin, these genes play a key role in the regulation of processes associated with the functioning of serotonergic neurons in MRNs of mice under the conditions of a chronic social conflict. Our findings support the recently advanced hypothesis that not only the CRH synthesized in the hypothalamus but also the CRH locally produced in CRH-containing raphe nuclei in the brainstem is important for stress adaptation. In addition, our results indicate that TRH locally produced in MRNs is also linked with serotonergic regulation of brain processes responsible for social behavior accompanying chronic positive or negative experiences under the conditions of chronic agonistic interactions. This may also be true in other cases of organismal adaptation to the effects of other stressors.

## Figures and Tables

**Figure 1 genes-12-01811-f001:**
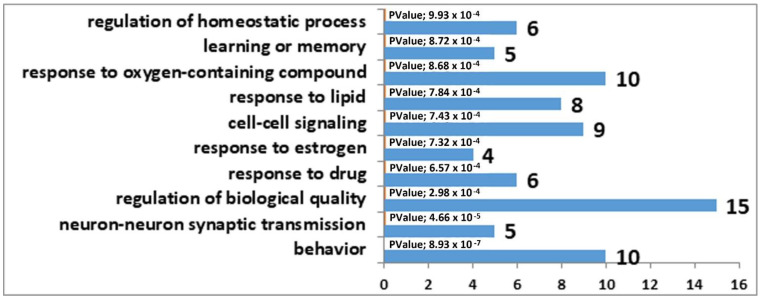
GO terms linked with the DEGs that correlated with *Tph2* expression in the MRN transcriptome analysis of the winners. The horizontal axis represents the number of genes.

**Figure 2 genes-12-01811-f002:**
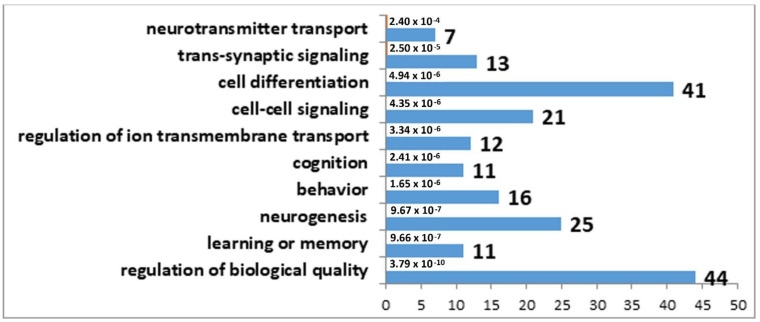
GO terms found to be enriched in the set of DEGs that correlated with *Tph2* expression in the MRN transcriptome analysis of the losers. The horizontal axis denotes the number of genes.

**Table 1 genes-12-01811-t001:** KEGG terms most significantly related to the DEGs in the MRNs of the winning mice compared to controls.

KEGG Term	Gene Count	*p* Value	Genes
Ribosome	13	1.18 × 10^−5^	*Rps5*, *Rplp1*, *Rpl34*, *Rplp0*, *Rpsa*, *Rpl10a*, *Rps16*, *Rps29*, *Rplp2*, *Rpl37*, *Uba52*, *Rps21*, *Rps12*
Cardiac muscle contraction	6	1.29 × 10^−2^	*Cacnb3*, *Uqcrq*, *Cox7a2*, *Uqcr11*, *Myh6*, *Cacng5*
Focal adhesion	10	1.31 × 10^−2^	*Prkcg*, *Lama5*, *Reln*, *Col24a1*, *Itga7*, *Pak6*, *Prkca*, *Parvb*, *Flnc*, *Mylk4*
Oxytocin signaling pathway	8	1.94 × 10^−2^	*Prkcg*, *Gucy1a3*, *Cacnb3*, *Kcnj12*, *Camk2a*, *Prkca*, *Cacng5*, *Mylk4*
GABAergic synapse	6	2.09 × 10^−2^	*Prkcg*, *Gabrb2*, *Gabra1*, *Gabra4*, *Gad2*, *Prkca*
Fatty acid biosynthesis	3	2.59 × 10^−2^	*Acsl1*, *Fasn*, *Acsbg1*
Serotonergic synapse	7	3.31 × 10^−2^	*Prkcg*, *Gabrb2*, *Tph2*, *Ddc*, *Prkca*, *Htr5b*, *Slc6a4*
Amphetamine addiction	5	3.34 × 10^−2^	*Prkcg*, *Ddc*, *Th*, *Camk2a*, *Prkca*
Tyrosine metabolism	4	3.35 × 10^−2^	*Ddc*, *Th*, *Mif*, *Dbh*
MAPK signaling pathway	10	3.93 × 10^−2^	*Prkcg*, *Cacnb3*, *Ptprr*, *Hspb1*, *Prkca*, *Flnc*, *Fgf13*, *Cacna1e*, *Cacng5*, *Cacna1g*
Calcium signaling pathway	8	4.58 × 10^−2^	*Prkcg*, *Chrm1*, *Camk2a*, *Prkca*, *Cacna1e*, *Htr5b*, *Cacna1g*, *Mylk4*
Endocrine and other factor-regulated calcium reabsorption	4	6.84 × 10^−2^	*Prkcg*, *Calb1*, *Ap2s1*, *Prkca*
ECM-receptor interaction	5	7.62 × 10^−2^	*Lama5*, *Reln*, *Col24a1*, *Itga7*, *Cd44*
Butanoate metabolism	3	8.59 × 10^−2^	*Hmgcl*, *Gad2*, *Aacs*
Morphine addiction	5	8.91 × 10^−2^	*Prkcg*, *Gabrb2*, *Gabra1*, *Gabra4*, *Prkca*
Dopaminergic synapse	6	9.72 × 10^−2^	*Prkcg*, *Ddc*, *Caly*, *Th*, *Camk2a*, *Prkca*
Long-term depression	4	9.93 × 10^−2^	*Prkcg*, *Gucy1a3*, *Crh*, *Prkca*

**Table 2 genes-12-01811-t002:** KEGG terms most significantly related to the DEGs in the MRNs of the losers (defeated mice) compared to controls.

KEGG Term	Gene Count	*p* Value	Genes
Cholinergic synapse	6	4.25 × 10^−3^	*Slc5a7*, *Chrnb4*, *Chrm1*, *Kcnj12*, *Prkca*, *Slc18a3*
Serotonergic synapse	6	8.16 × 10^−3^	*Tph2*, *Ddc*, *Htr3a*, *Prkca*, *Htr5b*, *Slc6a4*
MAPK signaling pathway	7	3.14 × 10^−2^	*Cacnb3*, *Hspb1*, *Prkca*, *Flnc*, *Fgf13*, *Cacna1e*, *Cacna1g*
Focal adhesion	6	4.63 × 10^−2^	*Reln*, *Col24a1*, *Pak6*, *Prkca*, *Parvb*, *Flnc*
Neuroactive ligand-receptor interaction	7	5.28 × 10^−2^	*Chrnb4*, *Chrm1*, *Gabra4*, *Aplnr*, *Gabre*, *Htr5b*, *Hcrtr1*
Type II diabetes mellitus	3	8.13 × 10^−2^	*Irs4*, *Cacna1e*, *Cacna1g*
Endocrine and other factor-regulated calcium reabsorption	3	8.70 × 10^−2^	*Kl*, *Calb1*, *Prkca*
Calcium signaling pathway	5	9.12 × 10^−2^	*Chrm1*, *Prkca*, *Cacna1e*, *Htr5b*, *Cacna1g*

**Table 3 genes-12-01811-t003:** Expression of DEGs dealing with the synthesis and transport of serotonin and DEGs encoding neuropeptides CRH and TRH.

**Gene Symbol**	**Gene ID**	**Expression in Controls, FPKM**	**Expression in Winners,** **FPKM**	**log_2_ (Fold Change) in Winners vs. Controls**	**q Value**	**Full Name**
**DEGs taking part in the synthesis and transport of serotonin**						
*Ddc*	13195	22.39	13.29	−0.75	4.64 × 10^−3^	dopa decarboxylase
*Slc6a4*	15567	19.58	7.20	−1.44	4.64 × 10^−3^	solute carrier family 6 (neurotransmitter transporter, serotonin), member 4
*Tph2*	216343	21.91	9.11	−1.27	4.64 × 10^−3^	tryptophan hydroxylase 2
***Crh* and *Trh* expression**						
*Crh*	12918	48.07	9.70	−2.31	4.64 × 10^−3^	corticotropin-releasing hormone
*Trh*	22044	30.06	4.28	−2.81	4.64 × 10^−3^	thyrotropin-releasing hormone
**Gene Symbol**	**Gene ID**	**Expression in Controls, FPKM**	**Expression in Losers,** **FPKM**	**log_2_ (Fold Change) in Losers vs. Controls**	**q Value**	**Full Name**
**DEGs taking part in the synthesis and transport of serotonin**						
*Ddc*	13195	22.23	8.80	−1.34	5.07 × 10^−3^	dopa decarboxylase
*Slc6a4*	15567	19.44	3.72	−2.39	5.07 × 10^−3^	solute carrier family 6 (neurotransmitter transporter, serotonin), member 4
*Tph2*	216343	21.75	4.40	−2.31	5.07 × 10^−3^	tryptophan hydroxylase 2
***Crh* and *Trh* expression**						
*Crh*	12918	47.71	5.76	−3.05	5.07 × 10^−3^	corticotropin-releasing hormone
*Trh*	22044	29.83	1.01	−4.88	5.07 × 10^−3^	thyrotropin-releasing hormone

**Table 4 genes-12-01811-t004:** Expression of DEGs coding for neurotransmitter and hormone receptors.

**DEGs Encoding Receptors for Neurotransmitters and Hormones in the MRNs of Winners versus Control Mice**							
**Gene Symbol**	**Gene ID**	**Expression in Controls, FPKM**	**Expression in Winners,** **FPKM**	**log_2_ (Fold Change) in Winners vs. Controls**	**q Value**	**Full Name**	**Neurotransmitters and Hormones**
*Chrm1*	12669	0.69	0.42	−0.73	1.98 × 10^−2^	cholinergic receptor, muscarinic 1, CNS	acetylcholine
*Gabra1*	14394	26.73	37.53	0.49	4.64 × 10^−3^	γ-aminobutyric acid (GABA) A receptor, subunit α 1	γ-aminobutyric acid
*Gabra4*	14397	3.74	2.33	−0.68	4.64 × 10^−3^	γ-aminobutyric acid (GABA) A receptor, subunit α 4	γ-aminobutyric acid
*Gabrb2*	14401	13.71	19.18	0.48	4.64 × 10^−3^	γ-aminobutyric acid (GABA) A receptor, subunit β 2	γ-aminobutyric acid
*Hcrtr1*	230777	6.22	3.95	−0.66	1.15 × 10^−2^	hypocretin (orexin) receptor 1	The encoded protein selectively binds the hypothalamic neuropeptide orexin A
*Htr5b*	15564	21.63	0.73	−4.89	4.64 × 10^−3^	5-hydroxytryptamine (serotonin) receptor 5B	serotonin
*Irs4*	16370	2.82	1.80	−0.65	8.27 × 10^−3^	insulin receptor substrate 4	insulin
*Nr2f2*	11819	5.06	8.17	0.69	4.64 × 10^−3^	nuclear receptor subfamily 2, group F, member 2	This gene encodes a member of the steroid thyroid hormone superfamily of nuclear receptors.
**DEGs encoding receptors for neurotransmitters and hormones in the MRNs of losers versus control mice**							
**gene symbol**	**gene ID**	**expression in controls, FPKM**	**expression in losers, FPKM**	**log_2_ (fold change) in losers vs. controls**	**q value**	**full name**	**neurotransmitters and hormones**
*Chrm1*	12669	0.69	0.41	−0.75	2.49 × 10^−2^	cholinergic receptor, muscarinic 1, CNS	acetylcholine
*Chrnb4*	108015	0.58	1.12	0.94	9.53 × 10^−3^	cholinergic receptor, nicotinic, β polypeptide 4	acetylcholine
*Gabra4*	14397	3.71	2.43	−0.61	3.50 × 10^−2^	γ-aminobutyric acid (GABA) A receptor, subunit α 4	γ-aminobutyric acid
*Gabre*	14404	2.37	1.48	−0.68	2.12 × 10^−2^	γ-aminobutyric acid (GABA) A receptor, subunit epsilon	γ-aminobutyric acid
*Hcrtr1*	230777	6.18	3.40	−0.86	5.07 × 10^−3^	hypocretin (orexin) receptor 1	the encoded protein selectively binds the hypothalamic neuropeptide orexin A
*Htr3a*	15561	1.99	0.97	−1.04	5.07 × 10^−3^	5-hydroxytryptamine (serotonin) receptor 3A	serotonin
*Htr5b*	15564	21.47	0.34	−5.99	5.07 × 10^−3^	5-hydroxytryptamine (serotonin) receptor 5B	serotonin
*Irs4*	16370	2.80	1.29	−1.12	5.07 × 10^−3^	insulin receptor substrate 4	insulin
*Nr2f2*	11819	5.03	8.80	0.81	5.07 × 10^−3^	nuclear receptor subfamily 2, group F, member 2	This gene encodes a member of the steroid thyroid hormone superfamily of nuclear receptors.

**Table 5 genes-12-01811-t005:** Correlation between the mRNA level of genes encoding proteins participating in the synthesis and transport of serotonin and the expression of genes coding for hormones or receptors of neurotransmitters or hormones.

**Winners/Control Mice Comparison**											
**Gene Symbol**	** *Chrm1* **	** *Crh* **	** *Gabra1* **	** *Gabra4* **	** *Gabrb2* **	** *Hcrtr1* **	** *Htr5b* **	** *Irs4* **	** *Nr2f2* **	** *Trh* **	
*Tph2*	0.040	**0.969**	−0.766	0.859	−0.569	0.671	**0.940**	**0.961**	−0.895	**0.987**	
*Ddc*	0.219	**0.983**	−0.826	**0.937**	−0.682	0.732	**0.978**	**0.970**	−**0.964**	**0.991**	
*Slc6a4*	0.090	**0.982**	−0.807	0.887	−0.624	0.720	**0.959**	**0.975**	−**0.926**	**0.996**	
**Losers/Control Mice Comparison**											
**Gene Symbol**	** *Chrm1* **	** *Chrnb4* **	** *Crh* **	** *Gabra4* **	** *Gabre* **	** *Hcrtr1* **	** *Htr3a* **	** *Htr5b* **	** *Irs4* **	** *Nr2f2* **	** *Trh* **
*Tph2*	0.419	−0.201	**0.994**	0.904	0.300	0.817	0.637	**0.993**	**0.973**	−**0.940**	**0.981**
*Ddc*	0.622	−0.424	**0.988**	**0.919**	0.473	0.904	0.711	**0.982**	**0.972**	−**0.988**	**0.967**
*Slc6a4*	0.427	−0.195	**0.992**	0.874	0.329	0.846	0.686	**0.985**	**0.958**	−**0.949**	**0.996**

Statistically significant correlations are highlighted in bold: blue, *p* < 0.01; red, *p* < 0.001.

**Table 6 genes-12-01811-t006:** The expression of genes encoding receptors for CRH or TRH or encoding the proteins that implement their inactivation as well as correlations.

**Gene Symbol**	**Expression in Controls, FPKM**	**Expression in Losers,** **FPKM**	**log_2_ (Fold Change) in Losers vs. Controls**	**q Value**	**Correlation with *Crh* Expression**
*Crh*	47.71	5.76	−3.05	5.07 × 10^−03^	1.000
*Crhbp*	9.97	4.60	−1.11	5.07 × 10^−03^	0.916
*Crhr1*	7.28	7.77	0.09	9.99 × 10^−01^	−0.690
					**correlation with *Trh* expression**
*Trh*	29.83	1.01	−4.88	5.07 × 10^−03^	1.000
*Trhr*	1.09	0.87	−0.33	9.99 × 10^−01^	0.585
*Trhr2*	1.18	1.33	0.18	9.99 × 10^−01^	−0.688
*Trhde*	1.78	1.36	−0.39	9.99 × 10^−01^	0.566
**Gene Symbol**	**Expression in Controls, FPKM**	**Expression in Winners,** **FPKM**	**log_2_ (Fold Change) in Winners vs. Controls**	**q Value**	**Correlation with *Crh* Expression**
*Crh*	48.07	9.70	−2.31	4.64 × 10^−03^	1.000
*Crhbp*	10.04	6.78	−0.57	3.84 × 10^−02^	0.915
*Crhr1*	7.33	7.05	−0.06	9.54 × 10^−01^	−0.697
					**correlation with *Trh* expression**
*Trh*	30.06	4.28	−2.81	4.64 × 10^−03^	1.000
*Trhr*	1.10	1.12	0.03	9.85 × 10^−01^	0.343
*Trhr2*	1.19	1.18	−0.01	9.96 × 10^−01^	−0.864
*Trhde*	1.79	1.69	−0.09	9.80 × 10^−01^	0.751

*Crhbp*: CRH-binding protein, which inactivates CRH; *Trhde*: thyrotropin-releasing-hormone–degrading enzyme.

## Data Availability

The RNA-Seq datasets are available in the European Nucleotide Archive (Accession No. PRJEB47635).

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
