# Peer review of "Correlation of Expression Changes between Genes Controlling 5-HT Synthesis and Genes Crh and Trh in the Midbrain Raphe Nuclei of Chronically Aggressive and Defeated Male Mice"

_genes, 2021, doi:10.3390/genes12111811_

Round 1
Reviewer 1 Report
Some minor points:
- Line 15, the phrase "social experience" could be modified by fighting experience. I consider, it is more specific for the aim of the study.
- Regarding title, I consider it is a little long. I could suggest: Correlation of expression genes controlling 5-HT synthesis and corticotrophin and thyrotropin genes in the raphe nuclei between aggressive and defeated mice in chronic agonistic interactions. The suggestion is entirely up to the authors' consideration.
Author Response
- Line 15, the phrase "social experience" could be modified by fighting experience. I consider, it is more specific for the aim of the study.
Answer: Thank you very much for your comment. We have made this change to the text of the manuscript.
- Regarding title, I consider it is a little long. I could suggest: Correlation of expression genes controlling 5-HT synthesis and corticotrophin and thyrotropin genes in the raphe nuclei between aggressive and defeated mice in chronic agonistic interactions. The suggestion is entirely up to the authors' consideration.
Answer: We agree with your comment and have made changes to the title, making it shorter. New title version is: Correlation of expression changes between genes controlling 5-HT synthesis and genes Crh and Trh in the midbrain raphe nuclei of chronically aggressive and defeated male mice
The authors are sincerely grateful to the referee for reading the manuscript, constructive comments and appreciation of our work.
Reviewer 2 Report
The study is timely and the well done.
The finding is interesting too.
In abstract, the authors addressed: The expression of the Tph2 gene encoding the rate-limiting enzyme of the serotonin synthesis pathway correlated with the expression of many genes, 31 of which were common be
tween aggressive and defeated mice and were downregulated in the MRNs of mice of both experimental groups.
For making their Introduction more comprehensive, they can consider also briefly addressing this issue in humans.
Here are some examples from the literature:
https://pubmed.ncbi.nlm.nih.gov/23965265/
https://pubmed.ncbi.nlm.nih.gov/32711172/
Author Response
The study is timely and the well done.
The finding is interesting too.
In abstract, the authors addressed: The expression of the Tph2 gene encoding the rate-limiting enzyme of the serotonin synthesis pathway correlated with the expression of many genes, 31 of which were common between aggressive and defeated mice and were downregulated in the MRNs of mice of both experimental groups.
For making their Introduction more comprehensive, they can consider also briefly addressing this issue in humans.
Here are some examples from the literature:
https://pubmed.ncbi.nlm.nih.gov/23965265/
https://pubmed.ncbi.nlm.nih.gov/32711172/
Answer: Thank you very much for this comment. We have made the following additions to the Introduction section to make it more comprehensive:
The effect of serotonin neurotransmission on social cognition and emotion management can be modulated by both genotype [11,12] and other factors, such as gender [13], age [14], as well as environmental factors [15,16], which makes it difficult to study the basic molecular genetic mechanisms that determine the synthesis of serotonin and its role in the formation of the phenotype on the human population.
- Lin, C.H.; Tseng, Y.L.; Huang, C.L.; Chang, Y.C.; Tsai, G.E.; Lane, H.Y. Synergistic effects of COMT and TPH2 on social cognition. Psychiatry 2013, 76, 273-294, doi:10.1521/psyc.2013.76.3.273.
- Zill, P.; Baghai, T.C.; Zwanzger, P.; Schule, C.; Eser, D.; Rupprecht, R.; Moller, H.J.; Bondy, B.; Ackenheil, M. SNP and haplotype analysis of a novel tryptophan hydroxylase isoform (TPH2) gene provide evidence for association with major depression. Mol Psychiatry 2004, 9, 1030-1036, doi:10.1038/sj.mp.4001525.
- Dykens, E.M.; Roof, E.; Bittel, D.; Butler, M.G. TPH2 G/T polymorphism is associated with hyperphagia, IQ, and internalizing problems in Prader-Willi syndrome. J Child Psychol Psychiatry 2011, 52, 580-587, doi:10.1111/j.1469-7610.2011.02365.x.
- Kataja, E.L.; Leppanen, J.M.; Kantojarvi, K.; Pelto, J.; Haikio, T.; Korja, R.; Nolvi, S.; Karlsson, H.; Paunio, T.; Karlsson, L. The role of TPH2 variant rs4570625 in shaping infant attention to social signals. Infant Behav Dev 2020, 60, 101471, doi:10.1016/j.infbeh.2020.101471.
- Xu, Z.; Reynolds, G.P.; Yuan, Y.; Shi, Y.; Pu, M.; Zhang, Z. TPH-2 Polymorphisms Interact with Early Life Stress to Influence Response to Treatment with Antidepressant Drugs. Int J Neuropsychopharmacol 2016, 19, doi:10.1093/ijnp/pyw070.
- Spagnolo, P.A.; Norato, G.; Maurer, C.W.; Goldman, D.; Hodgkinson, C.; Horovitz, S.; Hallett, M. Effects of TPH2 gene variation and childhood trauma on the clinical and circuit-level phenotype of functional movement disorders. J Neurol Neurosurg Psychiatry 2020, 91, 814-821, doi:10.1136/jnnp-2019-322636.
The authors are sincerely grateful to the reviewer for reading the manuscript, constructive comments and appreciation of our work.